# PCR Detection and Phylogenetic Analysis of Megalocytivirus Isolates in Farmed Giant Sea Perch *Lates calcarifer* in Southern Taiwan

**DOI:** 10.3390/v12060681

**Published:** 2020-06-24

**Authors:** Jia-Ming Tsai, Song-Lang Huang, Chung-Da Yang

**Affiliations:** 1Graduate Institute of Animal Vaccine Technology, College of Veterinary Medicine, National Pingtung University of Science and Technology, Pingtung 91201, Taiwan; jiamingtsainew@gmail.com; 2Pingtung County Animal Disease Control Center, Pingtung 90001, Taiwan; megayaya3@gmail.com; 3International Degree Program of Animal Vaccine Technology, International College, National Pingtung University of Science and Technology, Pingtung 91201, Taiwan

**Keywords:** iridovirus, megalocytivirus, red sea bream iridovirus (RSIV), infectious spleen and kidney necrosis virus (ISKNV), turbot reddish body iridovirus (TRBIV), giant sea perch iridovirus (GSPIV), phylogenetic analysis, polymerase chain reaction (PCR)

## Abstract

The *Megalocytivirus* genus includes three genotypes, red sea bream iridovirus (RSIV), infectious spleen and kidney necrosis virus (ISKNV), and turbot reddish body iridovirus (TRBIV), and has caused mass mortalities in various marine and freshwater fish species in East and Southeast Asia. Of the three genotypes, TRBIV-like megalocytivirus is not included in the World Organization for Animal Health (OIE)-reportable virus list because of its geographic restriction and narrow host range. In 2017, 39 cases of suspected iridovirus infection were isolated from fingerlings of giant sea perch (*Lates calcarifer*) cultured in southern Taiwan during megalocytivirus epizootics. Polymerase chain reaction (PCR) with different specific primer sets was undertaken to identify the causative agent. Our results revealed that 35 out of the 39 giant sea perch iridovirus (GSPIV) isolates were TRBIV-like megalocytiviruses. To further evaluate the genetic variation, the nucleotide sequences of major capsid protein (MCP) gene (1348 bp) from 12 of the 35 TRBIV-like megalocytivirus isolates were compared to those of other known. High nucleotide sequence identity showed that these 12 TRBIV-like GSPIV isolates are the same species. Phylogenetic analysis based on the MCP gene demonstrated that these 12 isolates belong to the clade II of TRBIV megalocytiviruses, and are distinct from RSIV and ISKNV. In conclusion, the GSPIV isolates belonging to TRBIV clade II megalocytiviruses have been introduced into Taiwan and caused a severe impact on the giant sea perch aquaculture industry.

## 1. Introduction

Epizootic iridoviral infection has threatened the aquaculture industry in the world, especially East and Southeast Asia, and has caused serious economic losses [1,2]. Iridoviruses are large DNA viruses, containing a linear, double-stranded DNA genome, displaying an icosahedral capsid morphology approximately 120~200 nm in diameter [3]. Based on the International Committee on Taxonomy of Viruses (ICTV), the family *Iridoviridae* consists of five genera: *Iridovirus*, *Chloriridovirus*, *Lymphocystivirus*, *Ranavirus* and *Megalocytivirus* [4,5]. Among these genera, *Megalocytivirus* is the most recently added genus [4,5]. Megalocytivirus-infected fish usually show lethargy, reduced appetite, severe anemia, petechiae of gills and enlarged abdomen [6]. In recent years, megalocytiviruses have drawn great attention because they usually induce lethal systemic infections that result in high morbidity and mortality in a wide range of freshwater and marine fish species [6]. 

The major capsid protein (MCP) gene has been employed to classify unknown megalocytivirus isolates due to its highly conservative DNA sequence [7,8]. Following the phylogenetic analyses based on the nucleotide sequences of the viral MCP gene, megalocytiviruses include three major genotypes or clusters [6] represented by (1) red sea bream iridovirus (RSIV), which was first reported in cultured red sea bream (*Pagrus major*) in Japan [9]; (2) infectious spleen and kidney necrosis virus (ISKNV), which was initially found in farmed mandarin fish (*Siniperca chuatsi*) raised for food in China [10,11]; (3) turbot reddish body iridovirus (TRBIV), which was first isolated from food turbot (*Scophthalmus maxius*) cultured in China [12] and South Korea [13,14]. So far, the RSIV genotype comprises those RSIV-like megalocytiviruses isolated from marine fish species in Japan, Korea, China, Taiwan and Southeast Asia [6,9,15,16]. On the other hand, up till now, the ISKNV genotype has included the ISKNV-like megalocytivirus isolates collected from mandarin fish in China [10,11,17,18] as well as food and ornamental fish cultured in Southeast Asia [7,8,19,20,21]. RSIV- and/or ISKNV-caused confirmed cases are widely detected in Southeast Asia, Taiwan and China [6]. Moreover, both RSIV- and ISKNV-induced iridoviral diseases are listed by the World Organization for Animal Health (OIE) as notifiable diseases in fish [6]. However, in contrast to RSIV- and ISKNV-like viruses, TRBIV-like megalocytiviruses are traditionally restricted to China [12,22] and South Korea [13,14,23] around the Yellow Sea in East Asia and mainly affect flatfishes, such as flounder and turbot [6]. Further genetic sequence investigations of more megalocytivirus isolates revealed that each genotype (RSIV, ISKNV and TRBIV) of the genus *Megalocytivirus* can be subdivided into two separate clades [6]. Those TRBIV isolates traditionally restricted in East Asia have been classified as the TRBIV clade I megalocytiviruses [24]. On the other hand, the newly found TRBIV clade II genotype has been so far discovered in freshwater ornamental fish [24,25] and marine-reared rock bream [8]. Therefore, much attention should be paid to further studies on the geographic distribution and the host range of TRBIV clade II megalocytiviruses. 

In Taiwan, surveillance studies on iridoviral outbreaks caused by megalocytiviruses have been undertaken [7,8,26]. Based on a significant phylogenetic analysis of iridoviruses collected from 2001 to 2009 in different cultured fish species in Taiwan, megalocytivirus infections have been found in fish cultured in Taiwan starting from 2005 [8]. In addition, most of these megalocytivirus isolates have been further identified as RSIV- and/or ISKNV-like megalocytiviruses and they have more affinity with the RSIV genotype than the ISKNV genotype [7]. However, TRBIV, the third genotype within the genus *Megalocytivirus*, is still very rare in Taiwan cultured fish [6]. 

According to the informative data from Food and Agriculture Organization of the United Nations (http://www.fao.org/fishery/culturedspecies/Lates_calcarifer/en), giant sea perch (*Lates calcarifer*) is one of the main consuming finfish species in Southeast Asia and can be cultured in both brackish and freshwater. The high quality of delicately-flavored white meat and high market demand make giant sea perch a very profitable fish for massive rearing in southern Taiwan. Generally, the breeders in southern Taiwan rear fingerlings of giant sea perch imported from Thailand. Recently, the megalocytivirus epizootics have occurred among giant sea perch farms in southern Taiwan. In 2017, a total of 39 iridoviruses were isolated from fingerlings of giant sea perch cultured in Kaohsiung and Pingtung, two important aquaculture geographic districts in southern Taiwan. In the present study, we undertook polymerase chain reaction (PCR) with different specific primer sets to screen these 39 giant sea perch iridovirus (GSPIV) isolates. Then, the major capsid protein (MCP) nucleotide sequences of these GSPIV isolates were cloned, sequenced and then compared to those of other known iridoviruses deposited in the GenBank to evaluate the genetic variation of these viruses. We found that the GSPIV isolates belonging to TRBIV clade II megalocytiviruses had been introduced into southern Taiwan to cause megalocytivirus epizootics among giant sea perch in 2017.

## 2. Materials and Methods

### 2.1. Fish and Virus

From April to October in 2017, iridoviral epizootics occurred in fingerlings of giant sea perch cultured in Kaohsiung and Pingtung, two geographic districts in southern Taiwan, and caused severe mortality up to 90% on affected farms. Thirty-nine cases of suspected iridovirus infection were isolated from these fingerlings, about 5~8 cm in length, collected from 39 different fish farms in Pingtung and Kaohsiung. Sources of these giant sea perch iridovirus (GSPIV) isolates used in the study are listed in Table 1. Among these isolates, GSPIV/TW38 was isolated from a giant sea perch fingerling which was just shipped to Pingtung from Thailand and immediately transported to our laboratory (Aquatic Animal Disease Laboratory, Graduate Institute of Animal Vaccine Technology, National Pingtung University of Science and Technology). 

All fingerlings we collected in the present study were commercially imported from Thailand. These diseased fish had reduced appetite, lethargy, deep body color, petechiae of gills, severe anemia and an enlarged abdomen. After euthanasia, their key internal characteristics included remarkably enlarged spleens and kidneys as well as reddish livers. These above-mentioned symptoms were consistent with those obvious pathological characteristics shown by megalocytivirus-infected fish. In addition, any apparent parasites or bacteria could not be recovered from tissues or organs of the diseased fish.

### 2.2. Viral DNA Extraction

The liver, spleen and kidney of euthanized fingerlings of giant sea perch were mixed together and homogenized in 500 µL of distilled water. After centrifugation at 3000 rpm for 10 min, the supernatant was collected and the total DNA containing viral DNA was then extracted and purified using the Blood and Tissue Genomic DNA Miniprep System (Viogene, New Taipei City, Taiwan) according to the manufacturer’s instructions. Finally, purified DNA used as a template for subsequent PCR was eluted in 200 μL of Elution buffer (10 mM Tris-HCl, 1 mM EDTA, pH 8.0) (Viogene, New Taipei City, Taiwan) and stored at −20 °C until use.

### 2.3. PCR Assays for the Identification and Differentiation of GSPIV Isolates from Giant Sea Perch

Since the symptoms we observed in giant sea perch fingerlings were similar to those of megalocytivirus-infected fish, PCR amplification of the specific gene fragments was undertaken in the present study to detect RSIV, ISKNV and TRBIV of the genus *Megalocytivirus*. All primer sets used for PCR detection in the present study were commercially synthesized (Tri-I Biotech, New Taipei City, Taiwan) and their sequences are listed in Table 2. The giant sea perch β-actin gene (GU188683.1), which can be amplified by PCR with the β-actin primer set (β-actin-F/β-actin-R) [27] shown in Table 2, was designed to serve as an internal control to assess the quality of each DNA sample extracted from the pooled liver, spleen and kidney of euthanized giant sea perch. Since RSIV and ISKNV are the two most commonly found megalocytiviruses in Southeast Asia, all 39 GSPIV isolates were initially screened and detected by PCR with the OIE-registered primer sets, 1-F/1-R (RSIV/ISKNV) and 4-F/4-R (RSIV), which have been designed for detection of RSIV and ISKNV [28,29]. In accordance with 2019 OIE-*Manual of Diagnostic Tests for Aquatic Animals*-*14/11/2019* (Chapter 2.3.8.) [30], the 1-F/1-R primer set can amplify the *Pst* I restriction fragment (570 bp) specific to both RSIV and ISKNV [28]. However, the DNA polymerase gene fragment (568 bp) only specific to RSIV, rather than ISKNV, can be amplified with the 4-F/4-R primer set [29]. Therefore, RSIV, rather than ISKNV, can be detected when both OIE-registered primer sets simultaneously amplify their target genes [28]. On the other hand, the detection result composed of positive 1-F/1-R PCR and negative 4-F/4-R PCR represents the presence of ISKNV infection [28,29]. Moreover, the published universal primer set (MCP-uni332-F3/MCP-uni1108-R8) specific for RSIV/ISKNV/TRBIV in the genus *Megalocytivirus* [6,31] was also employed to analyze 39 GSPIV isolates by PCR. Afterwards, those GSPIV isolates showing RSIV/ISKNV/TRBIV-positive were further differentiated their genotypes by PCR with three specific primer sets, previously proposed by Kurita and Nakajima [6], respectively, for RSIV (MCP-specR674-F4/MCP-specR888-R6), ISKNV (MCP-specI465-F3/MCP-specI879-R3) and TRBIV (MCP-specT37-F1/MCP-specT490-R1). In addition, in order to increase the detectable ability to screen and identify TRBIV megalocytiviruses, we designed a new reverse primer (hnMCP-specT342-R2) to work with the above-mentioned TRBIV forward primer (MCP-specT37-F1) for performing a hemi-nested (hn) PCR [32]. 

For each PCR reaction, 100 ng of extracted DNA (template) and 5~10 pmole of each specific primer were added into a total volume of 20 µL of AccuPower^®^ PCR PreMix (Bioneer, Daejeon, Korea) containing 250 µM dNTP, 10 mM Tris-HCl, 30 mM KCl, 1.5 mM MgCl_2_ and 1 U of Top DNA polymerase. The PCR reaction was performed in a Biometra TAdvanced Twin instrument (Labgene, Fribourg, Switzerland) based on the PCR conditions listed in Table 2. The amplified PCR products were then analyzed by electrophoresis in 2% agarose gels and stained with SafeVew^TM^ dye (BioPioneer, San Diego, CA, USA).

### 2.4. Viral MCP Gene Cloning and Sequencing

The MCP gene fragments (1348 bp) of the GSPIV isolates were amplified by PCR with the specific primer set, JM-MCP-F/JM-MCP-R, designed from the MCP nucleotide sequence of RBIV_Tp_45_08 (JF264352), a TRBIV isolate from rock bream [8]. Briefly, a 50-μL reaction mixture comprising 100 ng of template DNA, 10 pmole of each specific primer and AccuPower^®^ PCR PreMix (Bioneer, Daejeon, Korea) containing 250 µM dNTP, 10 mM Tris-HCl, 30 mM KCl, 1.5 mM MgCl_2_ and 1 U of Top DNA polymerase was subjected to an initial melting at 94 °C for 5 min followed by 30 cycles of amplification consisted of incubations at 95 °C for 30 s, 58 °C for 1 min 30 s and 72 °C for 1 min. A final extension step of 72 °C for 8 min was also carried out. The amplified PCR products (1348 bp) were cloned into yT&A cloning vector (Yeastern Biotech, New Taipei City, Taiwan) according to the manufacturer’s instructions and transformed into *Escherichia coli* DH-5α. Afterwards, the recombinant plasmid DNA was extracted separately from different clones and automatic sequencing was performed commercially using an ABI3730 automatic sequencer (Tri-I Biotech, New Taipei City, Taiwan).

### 2.5. Phylogenetic Analysis

Nucleotide sequence analyses of the MCP gene fragments from different GSPIV isolates were compared with those of other 57 known iridoviruses (Table 3) retrieved from the GenBank databases. Multiple nucleotide sequences were aligned using Clustal W [33]. Then, maximum likelihood (ML) phylogenetic tree based on the MCP nucleotide sequences was constructed using Molecular Evolutionary Genetics Analysis version X (MEGA X) to compare the relationships of GSPIV isolates to 57 known iridoviruses [34]. The phylogenetic tree was rooted against tiger frog virus (AF389451), a divergent iridovirus that has affinity with the genus *Ranavirus*.The Felsenstein bootstrap method with 1000 bootstrap replicates was used to test the reliability of the ML phylogenetic tree [35].

## 3. Results

### 3.1. PCR Identification of GSPIV Isolates from Giant Sea Perch

#### 3.1.1. The OIE-Registered Primer Sets

In 2017, a serious megalocytivirus epidemic occurred in giant sea perch cultured in southern Taiwan, including Pingtung and Kaohsiung. The infection had led to a systemic severe mortality up to 90% in giant sea perch farms. From April to October in 2017, we collected 39 moribund fingerlings of giant sea perch from 39 fish farms in Pingtung and Kaohsiung (Table 1) to identify the causative agent of the epizootics among giant sea perch. All DNA samples extracted from the pooled liver, spleen and kidney of euthanized giant sea perch were positive for the β-actin gene. Because RSIV and ISKNV are two megalocytiviruses that are widely found in Southeast Asian countries, PCR was first performed to identify the existence of RSIV and/or ISKNV using two OIE-registered primer sets, 1-F/1-R and 4-F/4-R (Table 2). According to our results shown in Table 4, neither the 1-F/1-R primer set nor the 4-F/4-R primer set could amplify PCR product from GSPIV/TW1~GSPIV/TW38. However, through both 1-F/1-R and 4-F/4-R primer sets, only GSPIV/TW39 showed a double-positive PCR reaction, thereby indicating that GSPIV/TW39 was a RSIV megalocytivirus. Thus, following examination with two OIE-registered primer sets, only No. 39 fish farm from which GSPIV/TW39 was collected was affected by RSIV infection.

#### 3.1.2. The Megalocytivirus Universal Primer Set

The low detection rate (1/39) done by OIE-registered primer sets, however, could not reflect the real severity of the megalocytivirus epidemic in giant sea perch in 2017. This further implied that there might be other megalocytiviruses taking part in the epidemic. In addition to OIE-registered primer sets, therefore, the universal primer set (MCP-uni332-F3/MCP-uni1108-R8) used to amplify the partial MCP fragment (777 bp) specific to megalocytiviral RSIV/ISKNV/TRBIV DNA (Table 2) was undertaken in the present study to examine these 39 GSPIV isolates. More surprisingly, when the universal primer set was used, PCR-positive reactions were found in 21 out of 39 GSPIV isolates (Table 4). It seemed that OIE-registered primer sets were unable to detect as more viruses as the universal primer set did.

#### 3.1.3. Primer Sets Respectively Specific to RSIV, ISKNV and TRBIV

To further determine the genotypes of these RSIV/ISKNV/TRBIV-positive GSPIV isolates, three different primer sets, previously proposed by Kurita and Nakajima, respectively, specific to RSIV (MCP-specR674-F4/MCP-specR888-R6), ISKNV (MCP-specI465-F3/MCP-specI879-R3) and TRBIV (MCP-specT37-F1/MCP-speT490-R1), were used to run PCR to discriminate between viruses (Table 2). As shown in Table 4, following PCR amplification with three specific primer sets, one RSIV-positive PCR, one ISKNV-positive PCR and nineteen TRBIV-positive PCR were found among these 21 RSIV/ISKNV/TRBIV-positive GSPIV isolates. Thus, based on the data derived from three specific primer sets, 48.7% (19/39) of GSPIV isolates collected from the megalocytivirus epizootics among giant sea perch in southern Taiwan in 2017 belonged to TRBIV-related viruses.

#### 3.1.4. The Hemi-Nested (hn) Primer Set

We then designed another TRBIV-specific primer set possessing a strong ability to detect a partial TRBIV MCP gene (306 bp) by performing hemi-nested (hn)-PCR (Table 2 and Figure 1). As shown in Table 4, 35 out of 39 GSPIV isolates were identified as TRBIV megalocytiviruses after hn-TRBIV PCR. Thus, the hn-TRBIV-positive rate reached to 90% (35/39 = 90%), which is higher than that (19/39 = 48.7%) detected by specific TRBIV PCR proposed by Kurita and Nakajima (Table 4). In other words, 16 GSPIV isolates (GSPIV/TW1, GSPIV/TW3, GSPIV/TW6, GSPIV/TW10, GSPIV/TW14, GSPIV/TW15, GSPIV/TW16, GSPIV/TW18, GSPIV/TW19, GSPIV/TW20, GSPIV/TW23, GSPIV/TW24, GSPIV/TW27, GSPIV/TW31, GSPIV/TW32, GSPIV/TW38) that could not be detectable using specific TRBIV PCR primer set proposed by Kurita and Nakajima were able to be revealed positive by hn-TRBIV PCR (Table 4 and Figure 1). Therefore, the detectable ability of the hn-TRBIV PCR primer set we designed in the present study is more powerful than that of the Kurita-and-Nakajima-proposed specific TRBIV PCR primer set.

### 3.2. Phylogenetic Analysis of GSPIV Isolates from Giant Sea Perch

Among these 35 hn-TRBIV PCR-positive GSPIV isolates, 12 isolates could be successfully cloned their MCP gene sequences (1348 bp) (Figure 1), thereby sequencing these cloned DNA fragments. The following 12 nucleotide sequence documents obtained in present study (Table 3) were deposited in the GenBank (accession No. (isolate)): MH237932 (GSPIV/TW2), MH237933 (GSPIV/TW4), MH237934 (GSPIV/TW5), MH237935 (GSPIV/TW7), MH237936 (GSPIV/TW8), MH237937 (GSPIV/TW11), MH237938 (GSPIV/TW17), MH237939 (GSPIV/TW22), MH237940 (GSPIV/TW25), MH237941 (GSPIV/TW29), MH237942 (GSPIV/TW30) and MH237943 (GSPIV/TW35). The MCP gene fragments of these 12 GSPIV isolates displayed 97.71~100% nucleotide sequence identity to each other (Table 5). In addition, these 12 GSPIV MCP gene sequences also showed 97.71~99.85% nucleotide sequence similarity (Table 5) to the MCP gene sequences of rock bream iridovirus RBIV_Tp_45_08 (JF264352), TRBIV case3 (KX354221), TRBIV case2 (KX354222) and TRBIV case1 (KX354223). Actually, since RBIV_Tp_45_08 (JF264352), TRBIV case3 (KX354221), TRBIV case2 (KX354222) and TRBIV case1 (KX354223) shared highly similar MCP sequences (Table 5), they had been identified as TRBIV megalocytiviruses in a previous study [24]. Therefore, the high extent of nucleotide sequence identity has indicated that these 12 GSPIV isolates are the same species as TRBIV megalocytiviruses represented by RBIV_Tp_45_08 (JF264352), TRBIV case3 (KX354221), TRBIV case2 (KX354222) and TRBIV case1 (KX354223).

To further examine the relationship between the 12 GSPIV isolates and previously published iridoviruses, we compared the MCP nucleotide sequences of 12 GSPIV isolates to those of other 57 known iridoviruses, including 56 megalocytiviruses and one ranavirus, tiger frog virus (AF389451), available in the GenBank (Table 3). All of the reference strains accessed from the GenBank and the 12 isolates of this study were divided into three main genotype clusters, RSIV genotype, ISKNV genotype and TRBIV genotype, separately (Figure 2). All of the 12 GSPIV isolates collected from Pingtung and Kaohsiung were grouped into the TRBIV-like genotype. Moreover, based on the phylogenetic analysis, 12 GSPIV isolates we collected in the present study and four known TRBIV-like megalocytiviruses, including RBIV_Tp_45_08 (JF264352), TRBIV case1 (KX354223), TRBIV case2 (KX354222) and TRBIV case3 (KX354221), were further put together into one group, the TRBIV clade II. Thus, these 12 GSPIV isolates (GSPIV/TW2, GSPIV/TW4, GSPIV/TW5, GSPIV/TW7, GSPIV/TW8, GSPIV/TW11, GSPIV/TW17, GSPIV/TW22, GSPIV/TW25, GSPIV/TW29, GSPIV/TW30 and GSPIV/TW35) belonged to the TRBIV clade II genotype.

## 4. Discussion

Megalocytiviruses have caused mass mortalities in various marine and freshwater fish species in East and Southeast Asia [6]. Recently, megalocytiviruses have been divided into three genotypes (RSIV, ISKNV and TRBIV) based on their MCP gene sequences [6]. Due to the traditional knowledge about the geographic restriction and the narrow host range, iridovirus epizootics resulting from TRBIV-like megalocytiviruses have been so far rare to be discovered in the global areas except East Asia [6,12,13,14,22,23]. Thus, TRBIV-like megalocytiviruses are not included in the OIE-reportable virus list [6]. In the present study, however, our results reveal that 35 out of the 39 GSPIV isolates we collected from the megalocytivirus epizootic in 2017 were identified as TRBIV-like megalocytiviruses by PCR (Table 4). Afterwards, 12 out of 35 TRBIV-like GSPIV isolates were used to clone their MCP gene fragments, and their high gene sequence identity revealed that these 12 TRBIV-like GSPIV isolates were the same species (Table 5). In addition, the phylogenetic analysis further demonstrated that 12 TRBIV-like GSPIV isolates were closely related to four known TRBIV clade II megalocytiviruses, including RBIV_Tp_45_08 (JF264352), TRBIV case1 (KX354223), TRBIV case2 (KX354222) and TRBIV case3 (KX354221), but distinct from RSIV and ISKNV megalocytiviruses (Figure 2). Thus, through PCR screening and identification, as well as genetic analysis based on the MCP gene, we have demonstrated that the GSPIV isolates belonging to TRBIV clade II megalocytiviruses have been found in the megalocytivirus epizootics among giant sea perch in southern Taiwan in 2017.

Giant sea perch aquaculture is a significant industry with high economic value in southern Taiwan, including Kaohsiung and Pingtung. Modern production technologies have further enhanced the intensification of giant sea perch aquaculture in Taiwan to meet the increasing food requirement for human consumption. However, such aquaculture intensification has been accompanied by the outbreaks of infectious diseases caused by viruses, especially megalocytiviruses [7,8]. To the best of our knowledge, TRBIV cases have never been found before in giant sea perch cultured in southern Taiwan. Therefore, based on PCR detection (Table 4) and phylogenetic analysis based on the MCP gene (Figure 2), this is the first report of TRBIV clade II megalocytivirus infection in farmed giant sea perch in Taiwan.

The amplification of a specific viral gene by PCR can be a valuable tool to discriminate and identify megalocytiviruses if the used primers possess sufficient specificity and sensitivity [36]. Since both RSIV and ISKNV megalocytiviruses are widely found in cultured fish reared in Southeast Asia [6], most researchers are used to screening and identifying suspected cases by PCR with the OIE-registered primer sets, 1-F/1-R and 4-F/4-R, which are designed to detect the presentence of RSIV and ISKNV [28,29]. Conceivably, TRBIV megalocytiviruses cannot be effectively detected by PCR with these OIE-registered primer sets [22,23]. According to our PCR results shown in Table 4, these OIE-registered primer sets were indeed unable to identify GSPIV isolates as TRBIV-like megalocytiviruses. However, further PCR detection with the universal primer set (positive rate: 21/39 = 53.8%) specific for RSIV/ISKNV/TRBIV pointed out the presence of TRBIV-like megalocytiviruses in the megalocytivirus epidemic in giant sea perch in 2017. Afterwards, three specific primer sets proposed by Kurita and Nakajima and the hn-TRBIV primer set were used to screen GSPIV isolates. The powerful PCR based on the hn-TRBIV primer set displayed a much stronger ability to identify 35 TRBIV-like megalocytiviruses from 39 GSPIV isolates, thereby resulting in the highest detection rate (35/39 = 90%), which is higher than that (19/39 = 48.7%) of the TRBIV-specific PCR suggested by Kurita and Nakajima [6]. Therefore, our results are consistent with those of previous studies, which have revealed that OIE-registered primer sets are unable to detect TRBIV-like megalocytiviruses [22,23]. Very few TRBIV cases found in Southeast Asia, especially in Taiwan, in the past, may be due to a lack of effective and trustworthy diagnostic methods to screen and confirm suspected cases [24]. Therefore, it seems that more powerful primer sets such as the hn-TRBIV PCR primer set we designed in the present study are necessary for TRBIV diagnosis in different fish species. In addition, recently, more sensitive real-time PCR has been employed to rapidly detect megalocytiviruses in fish [36], thereby achieving extensive detection and identification of megalocytiviruses in the future. However, among these 39 GSPIV isolates, GSPIV/TW12 and GSPIV/TW37 could not be detected by PCR with all primer sets used in the present study (Table 4). Such two isolates therefore require more analyses to screen and identify.

The transportation of infected fingerlings through international fish trade is an effective transmission pathway for the viral spread [37,38]. Giant sea perch is widely considered as a popular food fish species for massive rearing in southern Taiwan. Most fingerlings of giant sea perch are originally bred in sea water in Thailand and then shipped to southern Taiwan to continue to rear in fresh water for grow-out. In the present study, all fingerlings we collected for viral isolation were commercially imported from Thailand. Moreover, 90% of GSPIV isolates isolated from these fingerlings imported from Thailand were identified as TRBIV-like megalocytiviruses through hn-TRBIV PCR (Table 4). Of note, GSPIV/TW38 was isolated from a giant sea perch fingerling that was just shipped to Pingtung from Thailand and immediately transported to our laboratory. The positive hn-TRBIV PCR displayed by GSPIV/TW38 strongly suggests that the TRBIV infection in giant sea perch fingerlings occurred in Thailand. Apparently, based on our PCR screening data, most fingerlings may have been infected with TRBIV before importation to southern Taiwan. Nevertheless, more significant data are still needed to prove if such a viral transmission route through infected fingerlings imported from Thailand is possible.

Understanding of the host range and geographic distribution of a virus is important for the control and prevention of disease outbreaks. According to previous studies, the distribution of traditional TRBIV megalocytiviruses that usually infect food flatfish species is restricted to areas around the Yellow Sea in East Asia [6,12,13,14,22,23]. Recently, however, these traditional TRBIV megalocytiviruses have been further classified as the TRBIV clade I genotype, which is different from the TRBIV clade II genotype so far discovered in freshwater ornamental fish [24,25] and marine-reared rock bream [8]. Go et al. first reported and characterized the TRBIV clade II genotype based on the phylogenetic analyses of the MCP and ATPase gene sequences of iridoviruses from the archived materials of freshwater ornamental fish species (angelfish *Pterophyllum scalare* (KX354223), dwarf gourami *Trichogaster lalius* (KX354222) and oscar *Astronotus ocellatus* (KX354221)) in earlier outbreaks from the late 1980s through the early 1990s [24]. Since these archived ornamental fish were imported to Australia and North American from Southeast Asia, the TRBIV clade II megalocytiviruses may have spread around Southeast Asia for a long time starting from the late 1980s to the early 1990s [24]. Moreover, Koda et al. have sequenced two other TRBIV clade II megalocytiviruses, South American cichlid iridovirus (SACIV) and three spot gourami iridovirus (TSGIV), isolated from archival ornamental fish samples to provide the first complete genome sequences of the TRBIV clade II genotype [25]. In the same study, Koda and his coauthors have also expanded the geographic range of the TRBIV clade II genotype to include freshwater ornamental fish, including South American cichlids (oscar *Astronotus ocellatus* and keyhole cichlid *Cleithracara maronii*) and three spot gourami (*Trichopodus trichopterus*), traded in North America [25]. The most recent high-mortality outbreak initiated by a TRBIV clade II megalocytivirus, RBIV_Tp_45_08, was found in rock bream (*Oplegnathus fasciatus*) fingerlings imported into Taiwan from Korea in 2008 [8]. More importantly, in the present study, 12 GSPIV isolates that resulted in serious megalocytivirus epizootics in giant sea perch fingerlings imported into Taiwan from Thailand in 2017 have been found to belong to the TRBIV clade II genotype (Figure 2). Therefore, our finding and those recorded by others [8,24,25] have indicated that the TRBIV clade II megalocytiviruses have negatively impacted freshwater ornamental fish in Southeast Asia and North America as well as marine (rock bream) and freshwater (giant sea perch) food fish cultured in Southeast Asia. These data have also revealed that the viral transmission of TRBIV clade II megalocytiviruses is possible in both seawater and freshwater fish species [39]. Especially, the TRBIV clade II megalocytiviruses, RBIV_Tp_45_08 and GSPIV, have been introduced into Taiwan and caused dramatic economic losses on rearing both rock bream and giant sea perch. Apparently, like RSIV and ISKNV, the related TRBIV megalocytiviruses consisted of not only the TRBIV clade I but also the TRBIV clade II can also impose adverse economic impact due to the induction of dramatic reduction in production of freshwater ornamental and food fish species. In consideration of the severe influence on aquaculture industry, further studies are also needed to judge whether TRBIV should be considered, along with RSIV and ISKNV, as an OIE-reportable virus.

In summary, in the present study, GSPIV isolates belonging to the TRBIV clade II genotype have been demonstrated to be newly emerging viral fish pathogens in Taiwan. However, our findings in the present study are restricted to one species (giant sea perch) in one nation (Taiwan). Extensive monitoring of TRBIV clade II megalocytiviruses in different fish species in different areas should therefore be continued to improve the understanding of their host range and geographic distribution as well as their relationship to global epidemiology. On the other hand, the growth of GSPIV in cell culture is also needed to establish accurate viral quantification. Moreover, a proper viral challenge model to facilitate the development of effective preventive strategies, including vaccines and drugs, will be required in the future.

## 5. Conclusions

In the present study, 35 out of 39 GSPIV isolates collected from fingerlings of giant sea perch cultured in southern Taiwan during the megalocytivirus epizootics in 2017 have been identified as TRBIV-like megalocytiviruses by PCR. In addition, phylogenetic analysis of the MCP gene sequences from 12 TRBIV-like GSPIV isolates has shown that the 12 isolates belong to the TRBIV clade II megalocytiviruses and are distinct from RSIV and ISKNV. The GSPIV isolates belonging to TRBIV clade II megalocytiviruses have been introduced into Taiwan. Further studies are required to elucidate their local and global epidemiology as well as to develop effective preventive strategies.

## Figures and Tables

**Figure 1 viruses-12-00681-f001:**
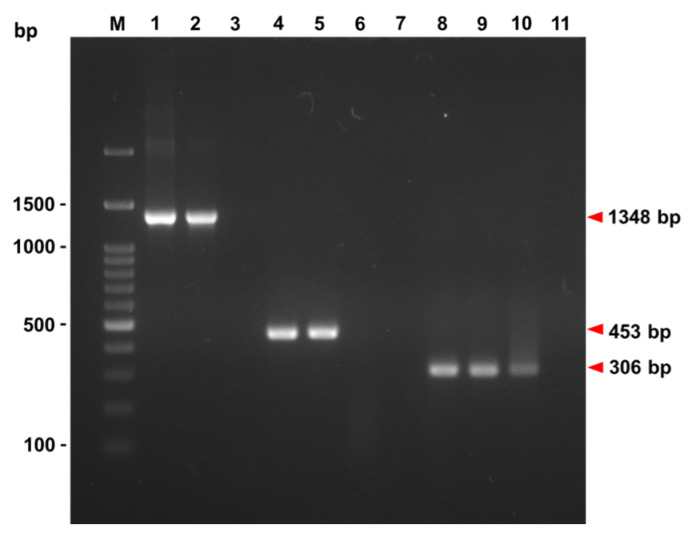
PCR with different primer sets was carried out to amplify GSPIV MCP gene fragments. The MCP gene fragment (1348 bp) from GSPIV/TW35 (Lane 2) was amplified using the JM-MCP–F/JM-MCP-R primer set for subsequent MCP gene cloning. GSPIV/TW35 (Lane 5) and GSPIV/TW38 (Lane 6) were screened by PCR with specific MCP-specT37-F1/MCP-specT490-R1 primer set proposed by Kurita and Nakajima to detect the TRBIV-specific MCP gene fragment (453 bp). The hn-TRBIV PCR with MCP-specT37- F1/hnMCP-specT342-R2 primer set was undertaken to detect another smaller TRBIV-specific MCP gene fragment (306 bp) from GSPIV/TW35 (Lane 9) and GSPIV/TW38 (Lane 10). Positive controls (20 ng of yT&A-TRBIV vector with cloned TRBIV MCP gene sequence) (Lanes 1, 4 and 8) and negative controls (no DNA template) (Lanes 3, 7 and 11) were also included. Standard DNA markers (lane M) are shown on both sides.

**Figure 2 viruses-12-00681-f002:**
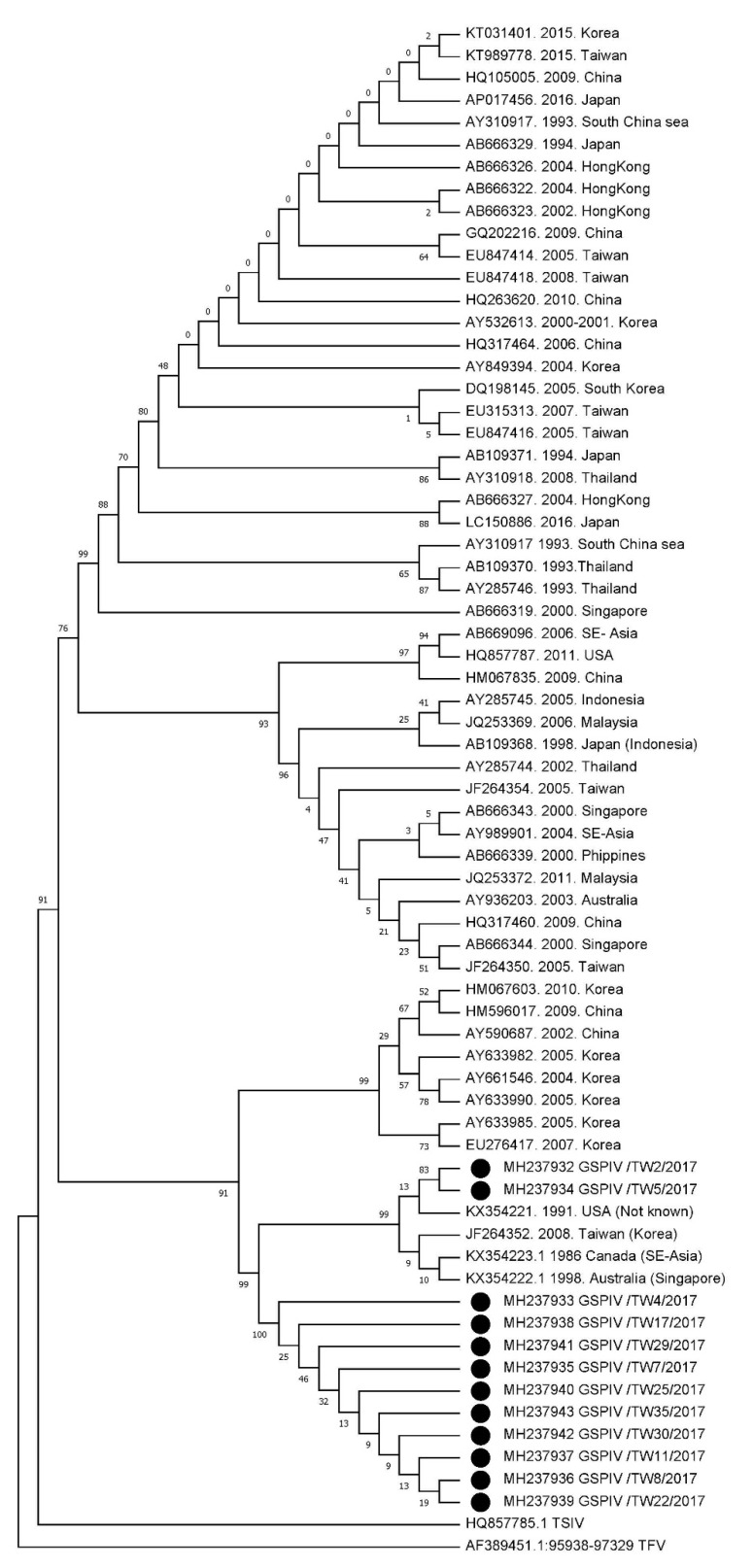
Phylogenetic analysis of magalocytivirus major capsid protein (MCP) gene. The nucleotide sequences (1348 bp) of the MCP gene from 12 GSPIV isolates (black dots) were compared with those of 57 other iridoviruses retrieved from NCBI GenBank. Virus names, accession numbers, year sampled, location of isolation and host of the retrieved iridoviruses are described in Table 3. The Maximum Likelihood tree was constructed by the neighbor-joining method using MEGA X [34]. The tree was rooted using MCP DNA sequences from tiger frog virus (AF389451.1), genus *Ranavirus*. The Felsenstein bootstrap method with 1000 bootstrap replicates was used to test the reliability of the Maximum Likelihood phylogenetic tree [35].

**Table 1 viruses-12-00681-t001:** Sources of iridoviruses isolated from cultured giant sea perch in southern Taiwan.

Isolate ^a^	Date (yr/mo) of Isolation	Fish Farm District ^b^
GSPIV/TW1	2017/04	PT
GSPIV/TW2	2017/04	PT
GSPIV/TW3	2017/04	PT
GSPIV/TW4	2017/05	K
GSPIV/TW5	2017/05	PT
GSPIV/TW6	2017/05	PT
GSPIV/TW7	2017/06	PT
GSPIV/TW8	2017/06	K
GSPIV/TW9	2017/06	K
GSPIV/TW10	2017/06	K
GSPIV/TW11	2017/06	K
GSPIV/TW12	2017/06	PT
GSPIV/TW13	2017/06	K
GSPIV/TW14	2017/07	PT
GSPIV/TW15	2017/07	PT
GSPIV/TW16	2017/07	PT
GSPIV/TW17	2017/07	PT
GSPIV/TW18	2017/08	PT
GSPIV/TW19	2017/08	PT
GSPIV/TW20	2017/08	PT
GSPIV/TW21	2017/08	PT
GSPIV/TW22	2017/08	PT
GSPIV/TW23	2017/08	PT
GSPIV/TW24	2017/08	PT
GSPIV/TW25	2017/09	PT
GSPIV/TW26	2017/09	PT
GSPIV/TW27	2017/09	PT
GSPIV/TW28	2017/09	PT
GSPIV/TW29	2017/09	PT
GSPIV/TW30	2017/09	PT
GSPIV/TW31	2017/09	PT
GSPIV/TW32	2017/09	PT
GSPIV/TW33	2017/09	PT
GSPIV/TW34	2017/09	PT
GSPIV/TW35	2017/09	PT
GSPIV/TW36	2017/09	PT
GSPIV/TW37	2017/09	PT
GSPIV/TW38	2017/09	PT
GSPIV/TW39	2017/10	PT

^a,b^ Giant sea perch iridovirus (GSPIV) isolates were isolated from giant sea perch cultured in Kaohsiung (K) and Pingtung (PT), two geographic districts in southern Taiwan (TW) in 2017.

**Table 2 viruses-12-00681-t002:** Primer sets used in the present study.

Purpose	Primer	Primer Sequence (5′-3′)	PCR Conditions	Product	Target Gene	Reference
**Detection**	**OIE RSIV/ISKNV**					
	1-F	CTCAAACACTCTGGCTCATC	94 °C, 5 min; 30 cycles: 95 °C, 30 s; 58 °C, 1 min; 72 °C, 1 min; 72 °C, 5 min	570 bp	*Pst* I fragment	[28]
	1-R	GCACCAACACATCTCCTATC				
	**OIE RSIV**					
	4-F	CGGGGGCAATGACGACTACA	94 °C, 5 min; 30 cycles: 95 °C, 30 s; 58 °C, 1 min; 72°C, 1 min; 72 °C, 5 min	568 bp	DNA Polymerase	[29]
	4-R	CCGCCTGTGCCTTTTCTGGA				
	**RSIV/ISKNV/TRBIV**					
	MCP- uni332- F3	AGGTGTCGGTGTCATTAACGACCT G	94 °C, 5 min; 30 cycles: 95 °C, 30 s; 58 °C, 1 min; 72 °C, 1 min; 72 °C, 5 min	777 bp	MCP	[6,31]
	MCP- uni1108- R8	TCTCAGGCATGCTGGGCGCAAAG				
	**RSIV**					
	MCP- specR674- F4	CCCGCACTGACCAACGTGTCC	94 °C, 5 min; 30 cycles: 95 °C, 30 s; 58 °C, 1 min; 72 °C, 1 min; 72 °C, 5 min	191 bp	MCP	[6]
	MCP- specR888- R6	CACAGGGTGACTGAACTCAGG TCG				
	**ISKNV**					
	MCP- specI465- F3	GGTGGCCGGCATCACCAACGG C	94 °C, 5 min; 30 cycles: 95 °C, 30 s; 58 °C, 1 min; 72 °C, 1 min; 72 °C, 5 min	413 bp	MCP	[6]
	MCP- specI879- R3	CACGGGGTGACTGAACCTG				
	**TRBIV**					
	MCP- specT37- F1	TTC ATC GAC ATC TCC GCT TTC	94 °C, 5 min; 30 cycles: 95 °C, 30 s; 58 °C, 1 min; 72 °C, 1 min; 72 °C, 5 min	453 bp	MCP	[6]
	MCP-specT490- R1	TST GAC CGT TGG TGA TAC CGG AG				
	**hn-TRBIV**					
	MCP- specT37- F1	TTC ATC GAC ATC TCC GCT TTC	94 °C, 5 min; 30 cycles: 95 °C, 30 s; 58 °C, 1 min; 72 °C, 1 min; 72 °C, 5 min	306 bp	MCP	Newly designed
	hnMCP-specT342-R2	CAC CGA CAC CTC CTC AAC C				
	**β−actin**					
	β−actin−F	TAC CAC CGG TAT CGT CAT GGA	94 °C, 5 min; 30 cycles: 95 °C, 30 s; 60 °C, 1 min; 72 °C, 1 min; 72 °C, 5 min	150 bp	β−actin	[27]
	β−actin−R	CCA CGC TCT GTC AGG ATC TTC				
**Cloning**	**JM-MCP**					
	JM-MCP-F	AGG TGC GAA CGT AAC CAG T	94 °C, 5 min; 30 cycles: 95 °C, 30 s; 58 °C, 1 min 30 s; 72 °C, 1 min; 72 °C, 8 min	1348 bp	MCP	Newly designed
	JM-MCP-R	TTA CAG GAT AGG GAA GCC TGC				

**Table 3 viruses-12-00681-t003:** GenBank accession numbers for the iridoviruses used in the major capsid protein (MCP) phylogenetic analysis.

Virus Name	Genus	MCP Accession No.	Country (origin)	Year	Host
Giant sea perch iridovirus (GSPIV/TW2/PT)	*Megalocytivirus*	MH237932	Taiwan	2017	Giant sea perch (*Lates calcarifer*)
Giant sea perch iridovirus (GSPIV/TW4/K)	*Megalocytivirus*	MH237933	Taiwan	2017	Giant sea perch (*Lates calcarifer*)
Giant sea perch iridovirus (GSPIV/TW5/PT)	*Megalocytivirus*	MH237934	Taiwan	2017	Giant sea perch (*Lates calcarifer*)
Giant sea perch iridovirus (GSPIV/TW7/PT)	*Megalocytivirus*	MH237935	Taiwan	2017	Giant sea perch (*Lates calcarifer*)
Giant sea perch iridovirus (GSPIV/TW8/K)	*Megalocytivirus*	MH237936	Taiwan	2017	Giant sea perch (*Lates calcarifer*)
Giant sea perch iridovirus (GSPIV/TW11/K)	*Megalocytivirus*	MH237937	Taiwan	2017	Giant sea perch (*Lates calcarifer*)
Giant sea perch iridovirus (GSPIV/TW17/PT)	*Megalocytivirus*	MH237938	Taiwan	2017	Giant sea perch (*Lates calcarifer*)
Giant sea perch iridovirus (GSPIV/TW22/PT)	*Megalocytivirus*	MH237939	Taiwan	2017	Giant sea perch (*Lates calcarifer*)
Giant sea perch iridovirus (GSPIV/TW25/PT)	*Megalocytivirus*	MH237940	Taiwan	2017	Giant sea perch (*Lates calcarifer*)
Giant sea perch iridovirus (GSPIV/TW29/PT)	*Megalocytivirus*	MH237941	Taiwan	2017	Giant sea perch (*Lates calcarifer*)
Giant sea perch iridovirus (GSPIV/TW30/PT)	*Megalocytivirus*	MH237942	Taiwan	2017	Giant sea perch (*Lates calcarifer*)
Giant sea perch iridovirus (GSPIV/TW35/PT)	*Megalocytivirus*	MH237943	Taiwan	2017	Giant sea perch (*Lates calcarifer*)
Rock bream iridovirus (RBIV-HD)	*Megalocytivirus*	KT031401	Korea	2015	Rock bream (*Oplegnathus fasciatus*)
Grouper iridovirus	*Megalocytivirus*	KT989778	Taiwan	2015	Grouper (*Epinephelus* sp.)
Rock bream iridovirus (RBIV-C1)	*Megalocytivirus*	HQ105005	China	2009	Rock bream (*Oplegnathus fasciatus*)
Red sea bream iridovirus (RIE12-1)	*Megalocytivirus*	AP017456	Japan	2016	Red sea bream (*Chrysophrys major*)
Sea bass iridovirus	*Megalocytivirus*	AY310917	China	1993	Sea bass (*Lateolabrax* sp)
Red sea bream iridovirus (RSIV-2)	*Megalocytivirus*	AB666329	Japan	1994	Greater amberjack (*Seriola dumerili*)
Red sea bream iridovirus (3GG1)	*Megalocytivirus*	AB666326	Hong Kong	2004	Orange-spotted grouper (*Epinephelus coioides*)
Red sea bream iridovirus (7GG)	*Megalocytivirus*	AB666322	Hong Kong	2004	Orange-spotted grouper (*Epinephelus coioides*)
Red sea bream iridovirus (GIG42)	*Megalocytivirus*	AB666323	Hong Kong	2002	Giant grouper (*Epinephelus lanceolatus*)
Spotted knifejaw iridovirus (SKIV-ZJ07)	*Megalocytivirus*	GQ202216	China	2009	Spotted knifejaw (*Oplegnathus punctatus*)
King grouper iridovirus (KGIV-05)	*Megalocytivirus*	EU847414	Taiwan	2005	Giant grouper (*Epinephelus lanceolatus*)
Barramundi perch iridovirus (BPIV-08)	*Megalocytivirus*	EU847418	Taiwan	2008	Barramundi perch (*Lates calcarifer*)
Stone flounder iridovirus (724)	*Megalocytivirus*	HQ263620	China	2010	Stone flounder (*Kareius bicoloratus*)
Rock bream iridovirus (SBIV-KOR-TY)	*Megalocytivirus*	AY532613	Korea	2000-2001	Sea bass (*Lateolabrax japonicus*)
Infectious spleen and kidney necrosis virus (ISKNV-HT)	*Megalocytivirus*	HQ317464	China	2006	Mandarin fish (*Siniperca chuatsi*)
Rock bream iridovirus (CNU-2)	*Megalocytivirus*	AY849394	South Korea	2004	Rock bream (*Oplegnathus fasciatus*)
Olive flounder iridovirus	*Megalocytivirus*	DQ198145	South Korea	2005	Bastard halibut (*Paralichthys olivaceus*)
Giant seaperch iridovirus (GSIV-K1)	*Megalocytivirus*	EU315313	Taiwan	2007	Giant sea perch (*Lates calcarifer*)
Barramundi perch iridovirus (BPIV-05)	*Megalocytivirus*	EU847416	Taiwan	2005	Barramundi perch (*Lates calcarifer*)
Red sea bream iridovirus	*Megalocytivirus*	AB109371	Japan	1994	Red sea bream (*Chrysophrys major*)
Red sea bream iridovirus	*Megalocytivirus*	AY310918	Japan/Thailand	2008	Red sea bream (*Chrysophrys major*)
Red sea bream iridovirus (6SB)	*Megalocytivirus*	AB666327	Hong Kong	2004	Yellowfin sea bream (*Acanthopagrus latus*)
Red seabream iridovirus (HyoDS-13)	*Megalocytivirus*	LC150886	Japan	2016	Devil stinger (*Inimicus japonicus*)
Sea bass iridovirus	*Megalocytivirus*	AY310917	China	1993	Sea bass (*Lateolabrax japonicus*)
Grouper sleepy disease iridovirus	*Megalocytivirus*	AB109370	Thailand	1993	Brown spotted grouper (*Epinephelus malabaricus*)
Grouper sleepy disease iridovirus	*Megalocytivirus*	AY285746	Thailand	1993	Brown spotted grouper (*Epinephelus malabaricus*)
Red seabream iridovirus (TGA12)	*Megalocytivirus*	AB666319	Singapore	2000	Brown-marbled grouper (*Epinephelus fuscoguttatus*)
Infectious spleen and kidney necrosis virus	*Megalocytivirus*	AB669096	Southeast Asia/Japan	2006	Banggai cardinalfish (*Pterapogon kauderni*)
Banggai cardinalfish iridovirus	*Megalocytivirus*	HQ857787	USA	2011	Banggai cardinalfish (*Pterapogon kauderni*)
Marble sleepy goby iridovirus	*Megalocytivirus*	HM067835	China	2009	Marble goby (*Oxyeleotris marmorata*)
African lampeye iridovirus	*Megalocytivirus*	AY285745	Indonesia	2005	African lampeye (*Aplocheilichthys normani*)
Megalocytivirus Sabah/RAA/2012 (HGIV67)	*Megalocytivirus*	JQ253369	Malaysia	2006	Barramundi cod (*Cromileptes altivelis*)
African lampeye iridovirus	*Megalocytivirus*	AB109368	Indonesia/Japan	1998	African lampeye (*Aplocheilichthys normani*)
Dwarf gourami iridovirus	*Megalocytivirus*	AY285744	Thailand	2002	Dwarf gourami (*Trichogaster lalius*)
Giant sea perch iridovirus (GSIV_Pt_843_05)	*Megalocytivirus*	JF264354	Taiwan	2005	Giant sea perch (*Lates calcarifer*)
Mullet iridovirus MA5/5	*Megalocytivirus*	AB666343	Singapore	2000	Fathead mullet (*Mugil cephalus*)
Dwarf gourami iridovirus	*Megalocytivirus*	AY989901	Southeast Asia	2004	Dwarf gourami (*Trichogaster lalius*)
Infectious spleen and kidney necrosis virus (lapulapu)	*Megalocytivirus*	AB666339	Philippines	2000	Orange-spotted grouper (*Epinephelus coioides*)
Megalocytivirus Sabah/RAA/2012 (OSGIV75)	*Megalocytivirus*	JQ253372	Malaysia	2011	Orange-spotted grouper (Epinephelus coioides)
Murray cod iridovirus	*Megalocytivirus*	AY936203	Australia	2003	Murray cod (*Maccullochella peeli*)
Infectious spleen and kidney necrosis virus (ISKNV-QY)	*Megalocytivirus*	HQ317460	China	2009	Mandarin fish (*Siniperca chuatsi*)
Dwarf gourami iridovirus DGA4/6K	*Megalocytivirus*	AB666344	Singapore	2000	Dwarf gourami (*Trichogaster lalius*)
Giant sea perch iridovirus (GSIV_Pt_836_05)	*Megalocytivirus*	JF264350	Taiwan	2005	Giant sea perch (*Lates calcarifer*)
Sea perch iridovirus (CH-1)	*Megalocytivirus*	HM067603	South Korea	2010	Sea perch (*Lateolabrax* sp.)
Turbot iridovirus isolate (R-603)	*Megalocytivirus*	HM596017	China	2009	Turbot (*Scophthalmus maximus*)
Turbot reddish body iridovirus	*Megalocytivirus*	AY590687	China	2002	Turbot (*Scophthalmus maximus*)
Korean flounder iridovirus (FLIV-MI)	*Megalocytivirus*	AY633982	Korea	2005	Olive flounder (*Paralichthys olivaceus*)
Olive flounder iridovirus	*Megalocytivirus*	AY661546	Korea	2004	Olive flounder (*Paralichthys olivaceus*)
Korean flounder iridovirus (FLIV-JJY)	*Megalocytivirus*	AY633990	Korea	2005	Olive flounder (*Paralichthys olivaceus*)
Korean flounder iridovirus (FLIV-WD2)	*Megalocytivirus*	AY633985	Korea	2005	Olive flounder (*Paralichthys olivaceus*)
Olive flounder iridovirus (OFLIV-1)	*Megalocytivirus*	EU276417	Korea	2007	Olive flounder (*Paralichthys olivaceus*)
Turbot reddish body iridovirus (case3)	*Megalocytivirus*	KX354221	USA	1991	Oscar (*Astronotus ocellatus*)
Rock bream iridovirus (RBIV_Tp_45_08)	*Megalocytivirus*	JF264352	Taiwan	2008	Rock bream (*Oplegnathus fasciatus*)
Turbot reddish body iridovirus (case1)	*Megalocytivirus*	KX354223	Canada/Southeast Asia	1986	Angelfish (*Pterophyllum scalare*)
Turbot reddish body iridovirus (case2)	*Megalocytivirus*	KX354222	Australia/ Singapore	1988	Dwarf gourami (*Trichogaster lalius*)
Threespine stickleback iridovirus	*Megalocytivirus*	HQ857785	Canada	2012	Threespine stickleback (*Gasterosteus aculeatus*)
Tiger frog virus	*Ranavirus*	AF389451	China	2002	Tiger frog (*Rana tigrina rugulosa*)

**Table 4 viruses-12-00681-t004:** Identification of giant sea perch iridovirus (GSPIV) isolates from diseased giant sea perch by PCR with different primer sets as shown in Table 2.

**Isolate**	**OIE Primer Sets ^a^**		**Universal Primer Set ^b^**	**Primer Sets Respectively Specific to ^c^**			**hn-TRBIV ^d^**
	**1-F/1-R**	**4-F/4-R**	**RSIV/ISKNV/TRBIV**	**RSIV**	**ISKNV**	**TRBIV**	
GSPIV/TW1	–	–	–	–	–	–	+
GSPIV/TW2	–	–	+	–	–	+	+
GSPIV/TW3	–	–	–	–	–	–	+
GSPIV/TW4	–	–	+	–	–	+	+
GSPIV/TW5	–	–	+	–	–	+	+
GSPIV/TW6	–	–	–	–	–	–	+
GSPIV/TW7	–	–	+	–	–	+	+
GSPIV/TW8	–	–	+	–	–	+	+
GSPIV/TW9	–	–	+	–	–	+	+
GSPIV/TW10	–	–	–	–	–	–	+
GSPIV/TW11	–	–	+	–	–	+	+
GSPIV/TW12	–	–	–	–	–	–	–
GSPIV/TW13	–	–	+	–	–	+	+
GSPIV/TW14	–	–	–	–	–	–	+
GSPIV/TW15	–	–	–	–	–	–	+
GSPIV/TW16	–	–	–	–	–	–	+
GSPIV/TW17	–	–	+	–	–	+	+
GSPIV/TW18	–	–	–	–	–	–	+
GSPIV/TW19	–	–	–	–	–	–	+
GSPIV/TW20	–	–	–	–	–	–	+
GSPIV/TW21	–	–	+	–	–	+	+
GSPIV/TW22	–	–	+	–	–	+	+
GSPIV/TW23	–	–	–	–	–	–	+
GSPIV/TW24	–	–	–	–	–	–	+
GSPIV/TW25	–	–	+	–	–	+	+
GSPIV/TW26	–	–	+	–	–	+	+
GSPIV/TW27	–	–	–	–	–	–	+
GSPIV/TW28	–	–	+	–	–	+	+
GSPIV/TW29	–	–	+	–	–	+	+
GSPIV/TW30	–	–	+	–	–	+	+
GSPIV/TW31	–	–	–	–	–	–	+
GSPIV/TW32	–	–	–	–	–	–	+
GSPIV/TW33	–	–	+	–	–	+	+
GSPIV/TW34	–	–	+	–	–	+	+
GSPIV/TW35	–	–	+	–	–	+	+
GSPIV/TW36	–	–	+	–	+	–	–
GSPIV/TW37	–	–	–	–	–	–	–
GSPIV/TW38	–	–	–	–	–	–	+
GSPIV/TW39	+	+	+	+	–	–	–

^a,b,c,d^ PCR reactions with World Organization for Animal Health (OIE) primer sets, universal primer set, three specific primer sets proposed by Kurita and Nakajima and hn-turbot reddish body iridovirus (TRBIV) primer set were used to screen 39 GSPIV isolates. (+) and (−) respectively indicate PCR-positive and PCR-negative reactions after PCR with different primer sets.

**Table 5 viruses-12-00681-t005:** Comparative analysis of nucleotide sequence identities (%) of the major capsid protein (MCP) gene among megalocytiviruses.

Virus Isolate ^a^	1	2	3	4	5	6	7	8	9	10	11	12	13	14	15	16
1. MH237932	100.00															
2. MH237933	98.01	100.00														
3. MH237934	99.85	98.01	100.00													
4. MH237935	97.86	99.85	97.86	100.00												
5. MH237936	97.78	99.77	97.78	99.62	100.00											
6. MH237937	97.78	99.77	97.78	99.62	99.54	100.00										
7. MH237938	98.01	100.00	98.01	99.85	99.77	99.77	100.00									
8. MH237939	97.71	99.69	97.71	99.54	99.46	99.46	99.69	100.00								
9. MH237940	97.86	99.85	97.86	99.69	99.62	99.62	99.85	99.54	100.00							
10. MH237941	97.94	99.92	97.94	99.77	99.69	99.69	99.92	99.62	99.77	100.00						
11. MH237942	97.86	99.85	97.86	99.69	99.62	99.62	99.85	99.54	99.69	99.77	100.00					
12. MH237943	97.86	99.85	97.86	99.69	99.62	99.62	99.85	99.54	99.69	99.77	99.69	100.00				
13. JF264352	99.85	98.01	99.85	97.86	97.78	97.78	98.01	97.71	97.86	97.94	97.86	97.86	100.00			
14. KX354221	99.85	98.01	99.85	97.86	97.78	97.78	98.01	97.71	97.86	97.94	97.86	97.86	99.93	100.00		
15. KX354222	99.85	98.01	99.85	97.86	97.78	97.78	98.01	97.71	97.86	97.94	97.86	97.86	99.93	100.00	100.00	
16. KX354223	99.85	98.01	99.85	97.86	97.78	97.78	98.01	97.71	97.86	97.94	97.86	97.86	99.93	100.00	100.00	100.00

^a^ The MCP gene sequences we cloned from 12 GSPIV isolates were compared with those of 4 iridoviruses (JF264352, KX354221, KX354222, KX354223). Homology is given as a percentage of nucleotide sequence identity. See Table 3 for descriptions of isolate abbreviations and GenBank accession numbers for MCP gene sequences.

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
