# Peer review of "PCR Detection and Phylogenetic Analysis of Megalocytivirus Isolates in Farmed Giant Sea Perch *Lates calcarifer* in Southern Taiwan"

_viruses, 2020, doi:10.3390/v12060681_

Round 1
Reviewer 1 Report
The manuscript by Jia-Ming Tsai and colleagues is based on identifying GSPIV isolates collected from fingerlings of giant sea perch cultured in southern Taiwan by PCR methodology and then a phylogenetic analysis of the MCP gene sequences was also evaluated. In conclusion, the phylogenetic analysis reveal that GSPIV isolates belonging to TRBIV clade II megalocytiviruses.
I found this article properly performed ,structured and and well written.
I only have some questions and minor comments for the authors.
Question:
1-It is well known that PCR is a perfect tool to identify viruses, in fact, in this paper authors use this method to evaluate infected samples. However, I think that real time PCR is even more rapid and sensitive than conventional PCR and it has also been used to detect megalocytiviruses (doi:10.1111/j.1365-2761.2011.01274.x). I wonder why authors did not use this technique? and If they think that if using the real time PCR maybe could be possible to identify more positive samples.
2-Apart from that, I would like to know if authors have analyzed some housekeeping gene in the PCR as internal control such as B-actin to assess the quality of the sample. I found a similar paper where they use this gene as control (doi: 10.1007/s00705-018-3834-6).
Minor comments :
-In the abstract define GSPIV because it is the first time that this initial appear in the manuscript.
- In the line 97 authors said :” Thirty-nine iridoviruses were isolated from these fingerlings”. This sentence is confused because it seems that you find 39 different iridoviruses and I think you want to indicate the number of infected individuals analyzed. Please check this sentence.
-In line 121 the elution buffer should include comercial reference.
-In line 172 Clustal W, authors should indicate the website or reference.
- In the discussion authors do not mention the result obtained with the universal primer set which they found 21 out of 39 PCR positive reaction. I think this result should be included.
Author Response
Comments and Suggestions for Authors
The manuscript by Jia-Ming Tsai and colleagues is based on identifying GSPIV isolates collected from fingerlings of giant sea perch cultured in southern Taiwan by PCR methodology and then a phylogenetic analysis of the MCP gene sequences was also evaluated. In conclusion, the phylogenetic analysis reveal that GSPIV isolates belonging to TRBIV clade II megalocytiviruses. I found this article properly performed, structured and well written.
I only have some questions and minor comments for the authors.
Question:
Point 1: It is well known that PCR is a perfect tool to identify viruses, in fact, in this paper authors use this method to evaluate infected samples. However, I think that real time PCR is even more rapid and sensitive than conventional PCR and it has also been used to detect megalocytiviruses doi:10.1111/j.1365-2761.2011.01274.x). I wonder why authors did not use this technique? and If they think that if using the real time PCR maybe could be possible to identify more positive samples.
Response 1:
Yes, we agree with Reviewer 1 on this point. Real-time PCR is more rapid and sensitive than conventional PCR. Conventional PCR, a convenient and cheap method for detecting viruses in fish, is the current detection strategy in our laboratory to diagnose megalocytiviruses. However, conventional PCR is laborious and slow due to its post amplification analysis, such as agarose gel electrophoresis and staining. Real-time PCR detection is usually systematically automated and does not need post amplification analysis. Since more sensitive and fast approaches are urgently needed to screen fish megalocytiviruses, we are planning to detect megalocytiviruses in food and ornamental fish in the future by running real-time PCR. In addition, the characteristic symptoms, including lethargy, reduced appetite, petechiae of gills, severe anemia and development of enlarged cells in the spleen, kidney and liver, usually observed in megalocytivirus-infected fish are also simultaneously checked to aid us to judge the megalocytivirus infection. We have incorporated the statement in the Discussion section (Lines 341-343) and also cited the following reference in the revised manuscript.
Gias, E.; Johnston, C.; Keeling, S.; Spence, R.P.; McDonald, W.L. Development of realtime PCR assays for detection of megalocytiviruses in imported ornamental fish. J Fish Dis 2011, 34, 609-618, doi:10.1111/j.1365-2761.2011.01274.x.
Point 2: Apart from that, I would like to know if authors have analyzed some housekeeping gene in the PCR as internal control such as B-actin to assess the quality of the sample. I found a similar paper where they use this gene as control (doi: 10.1007/s00705-018-3834-6).
Response 2:
In the present study, the DNA sample used as the PCR template was extracted from the pooled liver, spleen and kidney of euthanized giant sea perch. The -actin gene (GU188683.1) of giant sea perch was designed to serve as the internal control according to the previous study as follows. All DNA samples analyzed were positive for the -actin gene. Thus, the quality of each DNA sample was appropriate for running virus-specific PCR with different primer sets shown in Table 2. We have revised and incorporated the above information in section 2.3. (Lines 128-131) and section 3.1.1. (Lines 191-192) of our revised manuscript. We have also incorporated the -actin primer sequences into Table 2 and cited the following reference.
Paria, A.; Dong, J.; Babu, P.P.S.; Makesh, M.; Chaudhari, A.; Thirunavukkarasu, A.R.; Purushothaman, C.S.; Rajendran, K.V. Evaluation of candidate reference genes for quantitative expression studies in Asian seabass (Lates calcarifer) during ontogenesis and in tissues of healthy and infected fishes. Indian J Exp Biol 2016, 54, 597-605.
Minor comments:
Point 1: In the abstract define GSPIV because it is the first time that this initial appear in the manuscript.
Response 1:
We have incorporated “giant sea perch iridovirus (GSPIV)” in Abstract of the revised manuscript (Line 23).
Point 2: In the line 97 authors said: ”Thirty-nine iridoviruses were isolated from these fingerlings”. This sentence is confused because it seems that you find 39 different iridoviruses and I think you want to indicate the number of infected individuals analyzed. Please check this sentence.
Response 2:
Taking the advice of Reviewer 1, the sentence (Line 97) has been revised to “Thirty-nine cases of suspected iridovirus infection were isolated from these fingerlings, about 5~8 cm in length, collected from 39 different fish farms in Pingtung and Kaohsiung.”
Point 3: In line 121 the elution buffer should include commercial reference.
Response 3: We have included in the revised manuscript (Line 121).
Point 4: In line 172 Clustal W, authors should indicate the website or reference.
Response 4: We have incorporated the following reference in the revised manuscript (Line 176).
Thompson, J.D.; Higgins, D.G.; Gibson, T.J. CLUSTAL W: improving the sensitivity of progressive multiple sequence alignment through sequence weighting, position-specific gap penalties and weight matrix choice. Nucleic Acids Res 1994, 22, 4673-4680, doi:10.1093/nar/22.22.4673.
Point 5: In the discussion authors do not mention the result obtained with the universal primer set which they found 21 out of 39 PCR positive reaction. I think this result should be included.
Response 5:
Taking the advice of Reviewer 1, in the revised manuscript (Lines 329-332), we have inserted the following sentences: “However, further PCR detection with the universal primer set (positive rate: 21/39=53.8%) specific for RSIV/ISKNV/TRBIV pointed out the presence of TRBIV-like megalocytiviruses in the megalocytivirus epidemic in giant sea perch in 2017. Afterwards, three specific primer sets proposed by Kurita and Nakajima and the hn-TRBIV primer set were used to screen GSPIV isolates.”
Reviewer 2 Report
The authors describe a relatively unknown iridovirus in the genus megalocytivirus that causes a disease on giant sea perch in Taiwan. The virus was identified by PCR experiments and the authors developed primers that aided in the identification of the virus. The work is solid and the manuscript is well written. However, a good editor could improve on the writing.
Author Response
Comments and Suggestions for Authors
The authors describe a relatively unknown iridovirus in the genus megalocytivirus that causes a disease on giant sea perch in Taiwan. The virus was identified by PCR experiments and the authors developed primers that aided in the identification of the virus. The work is solid and the manuscript is well written. However, a good editor could improve on the writing.
Response:
Taking the advice of Reviewer 2, another round of proof-reading and editing has been performed to correct or modify other spelling and grammatical errors in the article. We thank Reviewer 2 for his effort on a constructive review of this study.